# Recent Advances in Colorimetric Tests for the Detection of Infectious Diseases and Antimicrobial Resistance

**DOI:** 10.3390/diagnostics13142427

**Published:** 2023-07-20

**Authors:** Cagla Celik, Gamze Kalin, Zekeriya Cetinkaya, Nilay Ildiz, Ismail Ocsoy

**Affiliations:** 1Pharmacy Services Program, Vocational School of Health Services, Hitit University, Corum 19000, Turkey; caglcel@gmail.com; 2Department of Infectious Diseases and Clinical Microbiology, Faculty of Medicine, Erciyes University, Kayseri 38039, Turkey; gamzekalinunuvar@erciyes.edu.tr; 3Kayseri State Hospital, Kayseri 38039, Turkey; zekeriyacetinkaya@hotmail.com.tr; 4Medical Imaging Department, Vocational School of Health Services, Bandırma Onyedi Eylul University, Bandirma 10200, Turkey; nilaygucluer@yahoo.com; 5Department of Analytical Chemistry, Faculty of Pharmacy, Erciyes University, Kayseri 38039, Turkey

**Keywords:** colorimetric test, bacteria, sensor, antimicrobial resistance

## Abstract

Diagnosis of infection-causing microorganisms with sensitive, rapid, selective and economical diagnostic tests is critical to start the right treatment. With these tests, the spread of infections can be prevented. In addition to that, the detection of antimicrobial resistance also makes a significant contribution to public health. In recent years, different types of diagnostic tests have been developed as alternatives to traditional diagnostic tests used in clinics. In particular, colorimetric tests, which minimize the need for an instrument, have advantages owing to their cost effectiveness, rapid response and naked-eye detection and practical use. In this review, we especially focused on pH indicators and nanomaterial-based colorimetric tests in detection of infection-causing microorganisms and antimicrobial resistance.

## 1. Introduction

Infectious diseases caused by various pathogens have become an emerging serious health problem. Thus, pathogens like foodborne, waterborne and hospital-acquired ones are among the high-risk groups in terms of both global economy and public health [1,2]. Various diseases caused by bacterial infectious agents are also cited as a cause of mortality and morbidity. In order to prevent the spread of infectious pathogens, accurate, sensitive and rapid diagnosis and the proper treatment methods should be emphasized [3,4,5].

Various traditional methods have been developed and have been in use in the field for the detection and identification of pathogenic bacteria. For instance, methods based on colony counting, culturing, ELISA (enzyme-linked immunosorbent assay) and PCR (polymerase chain reaction) have been actively used in clinics [3,4,5,6,7]. The enzymes are often adapted into the ELISA assay as signal enhancers owing to their substrate-specific catalytic activity. Three enzymes including alkaline phosphatase (ALP), horseradish peroxidase enzyme (HRP) and β-galactosidase are most commonly used in ELISA-based colorimetric immunoassays. The detection principle of ELISA simply relies on oxidation of or reduction in a substrate in the reaction environment. The changes in color type or intensity detected by the naked eye, a spectrometry device or a mobile application on smartphones usually proportionally increase in the target pathogen concentration [8].

The bio-components of microbial biosensors are usually microbial cells, but various microorganisms, tissues, cells and organelles are also included in this sensor group. The detection mechanism of some biosensors depends on amperometric, potentiometric or impedance electrochemical response [9]. These biosensors can be utilized to detect several metabolites such as H_2_, CO_2_, NH_3_ and organic acids found in aerobic organisms [9].

Phenotypic detection methods such as staining and culture have been used as a gold standard method for the pathogens, but a major disadvantage is that the result can be obtained at least in 48 h. Nevertheless, these methods do not provide sufficient information to report antibiotic resistance [10,11,12,13]. The delay in detection of serious infections such as sepsis can adversely influence treatment, which may induce death rate increases by 7.6% for every hour. Although the bacterial culture method is actively used, it must be performed by trained personnel and it cannot provide specific detection of bacteria owing to similarities in metabolic pathways and in phenotypic characteristics of bacteria. However, while molecular detection tests based on the analysis of genomic markers can result in significantly short time for microorganism detection, they often require specialized and expensive equipment and/or expert personnel and interpretation [10].

Conventional diagnostic kits present some challenges including isolation of the pathogen, trained personnel, specialized equipment, long time and high costs necessary to reach a diagnostic conclusion [14,15]. Therefore, there is still high demand for the development of novel methods that are economical, sensitive, specific, user-friendly, fast, and minimize the need for equipment. For these purposes, many different diagnostic tests and methods such as microfluidic devices, electrical sensors and DNA microarrays sensors have been designed [3,16,17,18,19,20,21,22,23]. Additionally, optical biosensors including photoluminescence, fluorescence and colorimetric based have been developed [24,25,26]. Among these, colorimetric biosensors have received considerable attention owing to fast and clear response. With these methods, the target pathogens in patient sample can be rapidly, easily, economically and selectively detected even with the naked eye by color change [27]. For instance, some colorimetric biosensors work based upon enzymatic reaction, and color change can be detected with the naked eye and with a device. Most of the time, colorimetric biosensors do not require an additional complicated equipment owing to visual response. For this purpose, various colorimetric tests have been developed for the diagnosis of pathogenic bacteria and antibiotic resistance.

Multidrug resistance in antibiotic treatment in bacteria is a serious threat that puts community healthcare at risk [28]. For instance, carbapenem resistance, especially in carbapenemase-producing bacteria, play a critical role in multidrug resistance [29,30]. Rapid and sensitive detection of resistant bacteria contributes to rational antibiotic use, protects public health and provides critical information for rapid implementation of epidemic-control measures [30]. Detection of carbapenem resistance may help to select the most suitable antibiotic treatment in the early term in the therapy of Gram-negative bacterial infections. In addition to that, rapid diagnosis of resistance provides advantages in many aspects such as shortening the length of antibiotic use, hospital stay, protecting patient immune system and reducing drug cost [31,32,33]. The active pathway in carbapenem resistance is based on the point that resistant microorganisms degrade the antibiotic by producing the carbapenemase enzyme. There are several analysis procedures for the rapid and sensitive detection of carbapenemase enzyme-release microorganisms through this mechanism, including UV spectrophotometric tests [34], matrix-assisted laser desorption ionization time-of-flight (MALDITOF) technological approaches [35,36,37] and molecular-based methods [38,39,40,41,42]. While the methods mentioned above perform with high sensitivity, specificity and accuracy, they require qualified personnel, are costly and involve time-consuming steps. Furthermore, molecular strategies may not catch carbapenemase-produced genes. As a solution to these problems, biochemical assays have been developed to confirm carbapenemase production and/or the presence of other mechanisms [43,44].

Nordmann and Poirel developed colorimetric tests called “NP Test” and introduced groundbreaking innovations in the sensitive and rapid colorimetric detection of antibiotic resistance. In these tests, a colorimetric response can be observed within 2 h, and it has the potential to be used directly on patient samples such as infected urine and blood cultures. For instance, the Carba NP tests [45] and their derivatives like Blue Carba [46] and β Carba [43] include biochemical reactions and economically and phenotypically verify carbapenemase enzyme release bacteria. The working principle of Carba NP tests is based on the release of carbapenemase enzymes during bacterial growth in bacterial culture (i.e., up to 24–48 h) which hydrolyze carbapenem group drugs such as imipenem or meropenem to produce an acidic environment. Then, a colorimetric response occurs due to the presence of an indicator in Carba NP tests. Thus, indicators such as phenol red or bromine thymol blue change color. Additionally, characteristic inhibitor agents of carbapenemase enzyme activity such as vaborbactam, avibactam or ethylenediamine tetra acetic acid can be included to increase sensitivity [43,47].

Recently, Celik et al. developed the red cabbage extract (RCE) incorporated media as a novel colorimetric test to diagnose Methicillin-resistant *Staphylococcus aureus* strains (MRSA). Anthocyanin is a natural pH indicator obtained from RCE which displays various colors in the different pH values, as shown in Figure 1A. For instance, anthocyanin molecules display pink, purple and blue–green colors in acidic, neutral and in alkaline conditions, respectively. Although susceptible bacteria are inhibited in the presence of antibiotics in this method, resistant bacteria continue to grow and release acidic volatile compounds. The color of the test medium changes from purple to pink in acidic liquid media, confirming the presence of MRSA [48].

Celik et al. also reported an anthocyanin-containing test in liquid and agar forms to distinguish MRSA and MRSE (Methicillin-resistant *Staphylococcus epidermidis*) based on pH change as presented in Figure 1B. Anthocyanin resistance tests have advantages over NO test in terms of sensitivity, detection time, cost effectiveness and biocompatibility of a pH indicator [49,51,52,53]. Researchers have the same strategy for colorimetric detection of urease-positive bacteria. For instance, *Helicobacter pylori* (*H. pylori*) colonized in the stomach secretes a specific urease enzyme which converts urea into ammonia (NH_3_) in in the reaction environment. The NH_3_ molecules make the reaction environment alkaline, and then the color of the reaction solution turns green from purple due to the deprotonation of anthocyanins shown in Figure 1C.

There are commercial products such as Pyloritek^®^, CLO test^®^, Hp One^®^, Pronto Dry^®^, and Hp Fast for the detection of *H. pylori*. While Pyloritek^®^, CLO test^®^ and Pronto Dry^®^ have phenol red as a pH indicator, Hp One^®^ and Hp Fast^®^ have bromothymol blue. Among these tests, only the CLO test^®^ offers the LOD value of 10^4^ in CFU/mL. However, the LOD value of the anthocyanin test decreases to 1 CFU/mL. In addition to the detection time of Pyloritek^®^, that of Hp One^®^ and Pronto Dry^®^ is about 1 h, while the CLO test^®^ and Hp Fast^®^ take 4–24 h for detection response. The anthocyanin test produces the colorimetric response in less than 1 h [50,54,55].

Due to the use of a biocompatible pH indicator, anthocyanin, incorporated in these colorimetric phenotypic tests, and a quite economical structure, these colorimetric phenotypic tests have a great potential to be practically used in clinics. Furthermore, anthocyanin-incorporated tests have also been utilized for the detection of urinary tract infection (UTI) pathogens [50,56]. Celik et al. also developed a test for the use in clinical laboratories to detect UTI. They designed a test to detect *Proteus mirabilis* (*P. mirabilis*) and *Klebsiella pneumonia* (*K. pneumonia*) causing UTI. These UTI bacterias exhibit urease enzyme activity. The urease enzyme hydrolyzes the urea to produce NH_3_ molecules. NH_3_ increases the pH of the test solution, in which the anthocyanin molecules in the test solution are deprotonated, then change color in response to the pH change. Commercial tests can detect pH changes using bromothymol blue, phenol red or bromocresol violet as pH indicators. These indicators are synthesized by chemical processes under laboratory conditions and may cause irritation in contact with eyes and skin. They have both corrosive and toxic properties and may cause irritation in the respiratory system when inhaled. For this reason, inhalation of the vapor or the gas form should be avoided. It is recommended to use protective materials to prevent contact with hands, face and eyes. Since anthocyanins, plant-derived pH indicators, are obtained from a plant consumed in daily life, there are no risks such as toxicity and irritation compared to other indicators. Anthocyanins obtained from red cabbage are frequently used in the coloring of food and fruit juices. In addition, regular consumption of food sources rich in anthocyanins is recommended due to their antioxidant effects. Red cabbage grows naturally, is easily accessible and is used as a plant source in antibiotic susceptibility tests due to its high anthocyanin content.

Color image processing techniques are also used for precise analysis of color change. A mobile application has been developed in which an image processing method can be applied in clinical laboratories. The mobile application includes interfaces that can be used directly by the technician. Thus, color changes are analyzed by preventing personal errors. In addition to that, a smartphone platform is printed on a 3D printer to minimize the factors affecting the analysis of the mobile application. With the smartphone platform and mobile application, the color change in the diagnostic test can be analyzed very precisely and quickly [56].

Common techniques such as culturing, microscopic urinalysis, ELISA testing, PCR, MALDI-TOF, FISH, and advanced light scattering are available for the detection of UTI. Although the analysis period of the culture method is quite long (48–72 h) and is used in clinical laboratories, it is the method used as the gold standard [57]. Although microscopic urine analysis is a rapid technique, it has poor sensitivity and specificity. In addition to that, it lacks the ability to detect antimicrobial resistance [58]. The ELISA test is a method that enables the indirect qualitative colorimetric detection of pathogens based on antigen–antibody combinations and can provide results in 2–3 h. However, a long process is required for the application of the method. In terms of sensitivity and specificity, it is evaluated as insufficient in the literature [59]. PCR is a method that provides results based on the amplification of specific genes (known to be specific to certain bacteria) from total genomic DNA extracted from urine samples, and is a sensitive and specific analysis method that can provide results in 5–6 h. Requirement of special probes for all pathogens, extensive and long preparation procedure and lack of quantitative data are the major disadvantages of the PCR method. In addition to that, the PCR method requires expensive devices and expert personnel [57,60]. In the detection of pathogens with the MALDI-TOF method, charged molecules are formed by ionization, separated according to the mass/charge ratio, and detected and measured using the TOF mass analyzer. Although it provides fast, precise and specific results, it is an expensive method. Sample preparation is difficult and interpretation of results is more complex than that of other methods [61,62]. In the FISH method, microscopic detection of microorganisms is performed by using fluorescently labeled nucleic acid probes hybridized to complementary targets. Although it is fast, sensitive and specific, its biggest disadvantage is the requirement of special probes for all pathogens [63]. In the forward light scattering method, bacterial growth is detected by forward light scattering based on changes, and the detection time is 90 min. It is cheap and requires a small amount of samples, and it is not able to identify the type of pathogen [64].

Nanoparticle (NPs)-based colorimetric biosensors including gold NPs (Au NPs), Ag/Au core-shell NPs, and quantum dots such as CdSe/ZnS have also been in use with biological elements [65,66,67,68,69,70]. Colorimetric detection methods based on NPs can be divided into three groups: (i) peroxidase-like activity of NPs, (ii) dispersion and aggregation of the NPs, and (iii) destabilization of the NPs.

## 2. Colorimetric Detection Based on Peroxidase-like Nature of NPs

Enzymes have been preferred as indication transducers in colorimetric detection methods due to their high sensitivity and specificity against certain substrates [71,72]. However, the use of enzymes in biosensors has been limited due to their high cost and short lifetime [73]. Various NPs with enzyme-like properties have been used as an alternative to enzymes [74,75,76]. Magnetic NPs, carbon nanotubes (CNTs), Au NP, graphene oxide (GO) and platinum (Pt) NP have been designed for the detection of pathogens via the peroxidase-like effect (Figure 2A). The OH^−^ radical can oxidize substrates and provide a colorimetric response. NPs conjugated with various targeting biomolecules such as DNA probes, DNA aptamers and antibodies are preferred in the specific and colorimetric detection of microorganisms via peroxidase-like activity [77,78,79,80,81]. For instance, antibody-conjugated Pt NP was developed for the colorimetric detection of Escherichia coli [78]. As a consequence of the peroxidase enzyme activity of Pt NP, the colorless TMB (3,3′,5,5′-Tetramethylbenzidine) molecule was oxidized and turned blue. In another study, Fe_3_O_4_ NP was functionalized with an aptamer for specific and colorimetric detection of *Salmonella typhimurium* (Figure 2B) [79]; similarly, colorless TMB was oxidized to a blue-color product [80,81]. The single-stranded DNA (ssDNA) aptamer was adsorbed on the positively charged surface of Fe_3_O_4_NPs; then, the aptamer was detached from Fe_3_O_4_ NPs in the presence of the pathogen and for binding to the pathogen. The naked Fe_3_O_4_ NPs oxidized TMB with peroxidase-like activity to form a blue-color TMBox (Figure 2B).

The GO and CNTs have attracted great interest for biosensor applications owing to their high conductive properties [82,83,84]. It was observed that GO, CNT, Au NPs@ GO and Au NPs@CNT act as Fenton reagents and show a highly sensitive colorimetric response in the presence of H_2_O_2_. For instance, the influenza A (H_3_N_2_) virus was sensitively detected using Au NPs@CNT nanocomposites (NCs). The haemagglutinin (HA) monoclonal antibodies were bound to Au NPs@CNT NC to selectively recognize influenza A (H_3_N_2_) and show peroxidase-like activity (Figure 2C) [85]. In another study, norovirus was sensitively and specifically detected by enhancing the peroxidase enzymatic activity of Au NPs@ GO NCs (Figure 2C) [86].

**Figure 2 diagnostics-13-02427-f002:**
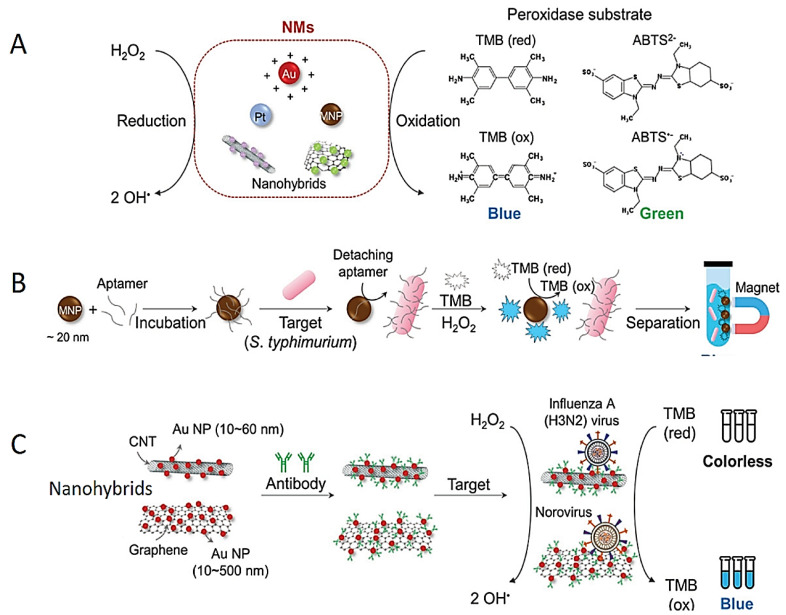
Schematic designs of colorimetric tests based on peroxidase-like activities of the NPs. (**A**) Scheme of peroxidase-like activity of different NPs. (**B**) Scheme of *S. Typhimurium* detection test using aptamer conjugated Fe_3_O_4_ NPs. (**C**) Colorimetric detection of influenza (H_3_N_2_) virus using Au NPs@CNTs NCs. Reprinted from Ref. [87].

## 3. Colorimetric Detection Based on Dispersion and Aggregation of NPs

Because of the extraordinary optic properties of plasmonic NPs (especially Ag and Au NPs), the direct use of these NPs in colorimetric diagnostic tests has also been presented as a new approach. The changes on the surface of the NPs may lead to dispersion or aggregation, both of which may induce color changes [66]. These color changes are observed by naked eye and spectrophotometer. The color change in NPs provides a practical and easy platform in colorimetric tests for the detection of pathogens (Figure 3A). The Ag and Au NPs are most frequently preferred for monitoring color changes due to their unique plasmonic properties. For instance, while colloidal Au NPs in solutions produce a red color, a blue–purple color appears due to the aggregation of Au NPs in the solution. Similarly, the Ag NP solution is usually of a yellow color, and when the particles are aggregated, the color of the Ag NP solution becomes red.

The surface of these NPs can be conjugated with RNA, DNA and/or antibodies for a selective detection of targets (Figure 3A). For instance, pathogen-specific antibodies are conjugated to the surface of Au NPs; then, the Au NPs come together on the surface of the target pathogen. Thus, color change occurs due to the aggregation of Au NPs, and colorimetric response is observed by the naked eye [74,88,89,90,91]. Furthermore, nanoliposome NPs have been used as a signal amplification tool for the detection of foodborne pathogens (Figure 3B) [92]. Mouse monoclonal antibody-conjugated nanoliposome NPs detect certain pathogens by containing cysteine molecules that induce the aggregation of Au NPs. Mouse monoclonal antibodies molecularly recognize the pathogens such as *E. coli O157:H7*, *S. typhimurium* and *Listeria monocytogenes* (*L. monocytogenes*); subsequently, polyclonal antibodies bind to target pathogens. However, this multi-step process is a serious disadvantage for industrial production. The complex and time-consuming preparation process limits its applicability to commercial kits. To address this issue, the use of unmodified NPs was aimed to facilitate the commercialization of colorimetric assays [3]. The thiol-Au binding method developed for this purpose has been widely used as a DNA conjugation method for enhanced binding potency [93,94]. Therefore, unmodified DNA adsorption on unmodified Au NPs has been used for the detection of various pathogens [95]. For instance, the DNA of Cyprinid herpesvirus-3 pathogen was successfully identified based on the aggregation using citrate-stabilized Au NP (Figure 3C) [96].

Furthermore, single-stranded and double-stranded DNA molecules can be distinguished by producing different responses in the colorimetric test in the presence of NaCl because of their distinct electrostatic interactions [97]. Single-stranded DNA (ssDNA) can be adsorbed on the surface of Au NPs, and no aggregation occurs. However, the double-stranded DNA (dsDNA) cannot be adsorbed on the surface of Au NPs owing to the blockage of pi electrons. Based on this, in the presence of herpesvirus-3 DNA, the ssDNA probe reacts with herpesvirus-3 DNA to form the dsDNA structure, then leading to the assembling of Au NPs, providing a colorimetric result. A similar method was used to detect double-stranded DNA of the *S. typhimurium* pathogen using Au NPs [98].

In another study, pyrrolidinyl peptide nucleic acid known as acpc PNA was bonded on (2S)-aminocyclopentane-(1S)-carboxylic acid to detect *Mycobacterium tuberculosis*, human papillomavirus and Middle East respiratory syndrome coronavirus in the presence of Ag NPs (Figure 3D) [99]. The acpc PNA probe induced the aggregation of Ag NPs, leading to the color changing from yellow to red due to electrostatic interactions. When PNAs bind to target DNAs, Ag NPs are stabilized in the test environment. There exists a mention that this mechanism has been used in the improvement of paper-based analytical instruments owing to its easy integration, affordability and portability [100,101,102]. Carbapenem-resistant bacterial strains including *Rhodopseudomonas*, *Acetobacter aceti*, *Staphylococcus* and *E. coli* were detected colorimetrically using unmodified Au NPs and Ag NPs [103].

## 4. Colorimetric Detection Based on Destabilization of NPs

Various tests for the colorimetric identification of pathogens have also been developed based on the modification and stabilization of the NPs. The most frequently used NPs are polydiacetylene (PDA)-based structures with interesting chromatic features [91,104]. Blue-color PDA structures are synthesized by 1,4 photopolymerization of self-assembled diacetylene monomers. The specific aptamer-, antibody- and/or peptide-conjugated PDA structures for the detection of pathogens are shown in Figure 4A. For instance, the color of the peptide-bound PDA vesicle system changes from blue to red; this was used for the identification of the influenza H1N1 virus (Figure 4B) [105].

In another study, the PDA/sphingomyelin (SPH)/cholesterol (CHO)/lysine structure was used as a colorimetric test for the identification of *Salmonella choleraesuis* in chicken (Figure 4B) [106]. Although highly sensitive detection could be achieved using the PDA vesicle in this study, long incubation time (almost 48 h) was needed.

## 5. Conclusions and Future Perspectives

This review presents recent studies on the identification of various pathogens by different types of colorimetric diagnostic tests. NPs with unique electrical, optical and catalytic features have been used for specific and sensitive detection of microorganisms. Nevertheless, there are still some challenges for NP-based colorimetric diagnostic tests in terms of stability, need of expertise and cost.

There is a high demand for the development of economical, rapid and sensitive colorimetric diagnostic tests. In particular, metals such as Ag, Au and Pt NPs have superior optical, catalytic and electrical properties but may not be economical enough for industrial production and large-scale applications [107,108]. In addition to that, controlling the size and shape of NPs is a crucial point for sensitivity and selectivity, because chemical, physical and biological properties including catalytic properties are directly dependent on the size and shape of NPs [67,85,86,109,110,111]. For instance, smaller-size Au NPs or Au NPs@GO NCs have high surface area and consequently show increase in peroxidase-like activities [86].

Various pH-indicator-based phenotypic and colorimetric tests have been developed for rapid, economical, sensitive and accurate detection of infection-causing microorganisms and antimicrobial resistance with the naked eye. While these colorimetric tests do not require any complicated and expensive devices, and even expert personnel to use, they are considered to be portable and point-of-care (POC) tests.

## Figures and Tables

**Figure 1 diagnostics-13-02427-f001:**
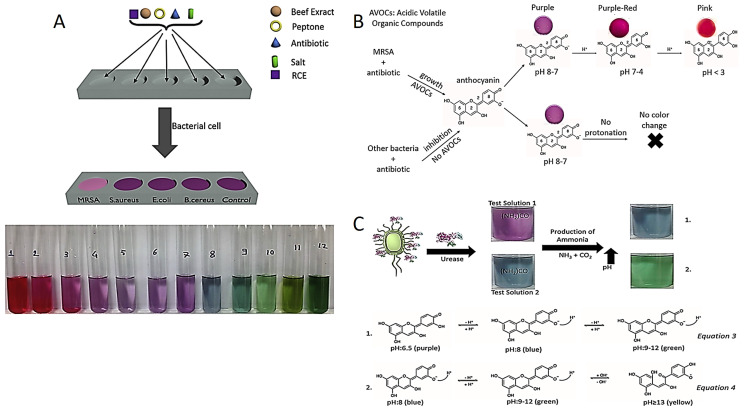
(**A**) MRSA diagnostic test components and anthocyanin pH-dependent color change. Reprinted with permission from Ref. [48]. Copyright 2023 Elsevier. (**B**) Color change in anthocyanin-containing agar in the presence of MRSA. Reprinted with permission from Ref. [49]. Copyright 2023 Elsevier. (**C**) Detection of *H. pylori* by colorimetric tests prepared at two different pH and equations for color change mechanisms. Reprinted from Ref. [50].

**Figure 3 diagnostics-13-02427-f003:**
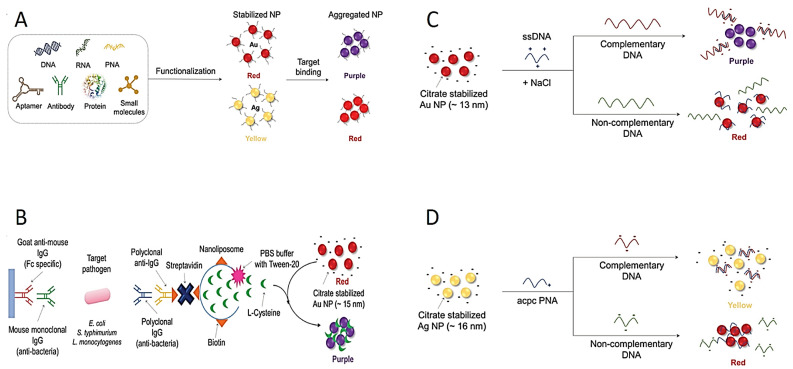
Schematic explanation of colorimetric diagnostic tests based on dispersion and aggregation of NPs. (**A**) Scheme of aggregation of different-molecule-integrated NPs. (**B**) Signal-amplified detection of pathogens using protein functionalized nanoliposomes. (**C**) Schematic design of detection of the virus DNA using aggregation of Au NP in the existence of dsDNA. (**D**) Schematic design of detection of the Middle East respiratory syndrome coronavirus DNA using redispersion–aggregation mechanism of Ag NP. Reprinted from Ref. [87].

**Figure 4 diagnostics-13-02427-f004:**
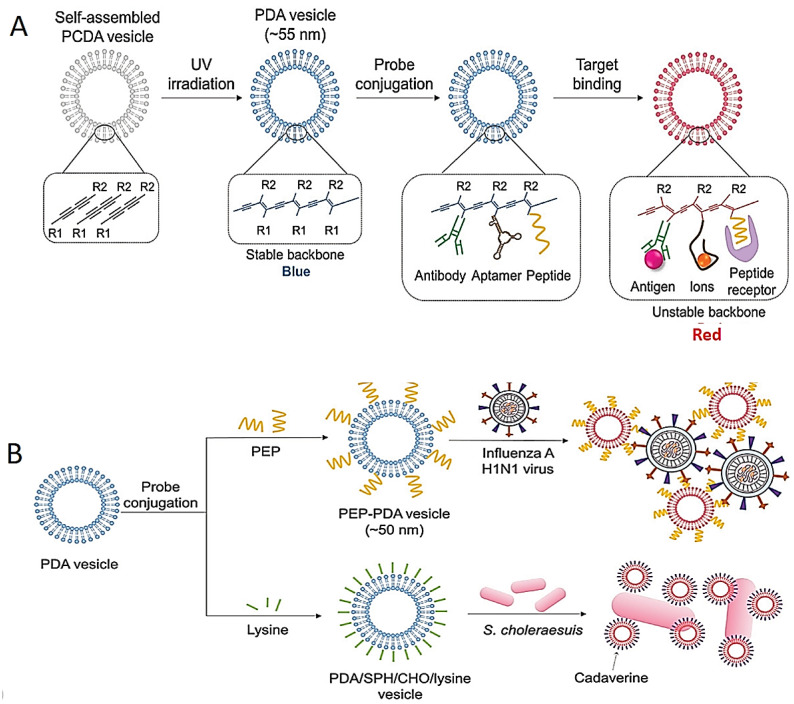
Schematic images of colorimetric test based on destabilization of NP structures. (**A**) Image of color-changing of PDA structures by their structure transition on target interactions. (**B**) Schematic illustrations of identification of influenza H1N1 virus and *Salmonella choleraesuis* with PEP–PDA and PDA/SPH/CHO/lysine structure, respectively. Reprinted from Ref. [87].

## Data Availability

Not applicable.

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
