# Peer review of "Recent Advances in Colorimetric Tests for the Detection of Infectious Diseases and Antimicrobial Resistance"

_diagnostics, 2023, doi:10.3390/diagnostics13142427_

Round 1

Reviewer 1 Report

The review paper titled "Recent Advances in Colorimetric Tests for the Detection of Infectious Diseases and Antimicrobial Resistance" reported one of the techniques used to detect microorganisms causing infection. This review provides a sensitive, rapid, and selective diagnosis of microorganisms with calorimetric tests, especially pH indicators and nanomaterial-based ones. It emphasizes the potential of pH indicators and nanomaterial-based calorimetric tests. The manuscript can be publishable in the “Diagnostics” after some revisions.

1.              The figures in the manuscript are seen as blurry. The image quality of the figures in the manuscript should be clarified.

2.              The subject transitions in the manuscript should be written more fluently.

3.              The words such as "designedfor", "biosensorsdepends" "beenused", "agold", " throughshowing", “Fe3O4NPsoxidized”,... is written incorrectly. There are too many typos in the manuscript. The typos in the manuscript should be corrected.

4.              On page 1, lines 29-30, appropriate references should be added to the sentence in the introduction section.

5.              Figure 1 (especially Figure 1A), should be presented more clearly with necessary and informative information.

6.              Explanations of abbreviations like “TMB molecule, TMBox, Ag” should be added in the manuscript.

Quality of English Language is good

Author Response

A response to the comments made by the editors the reviewer

Submission ID: diagnostics-2445497

Title: Recent Advances in Colorimetric Tests for the Detection of Infectious
Diseases and Antimicrobial Resistance

Journal: Diagnostics

Reviewer #1:

The review paper titled "Recent Advances in Colorimetric Tests for the Detection of Infectious Diseases and Antimicrobial Resistance" reported one of the techniques used to detect microorganisms causing infection. This review provides a sensitive, rapid, and selective diagnosis of microorganisms with calorimetric tests, especially pH indicators and nanomaterial-based ones. It emphasizes the potential of pH indicators and nanomaterial-based calorimetric tests. The manuscript can be publishable in the “Diagnostics” after some revisions.

Response to Reviewer #1: We are very grateful to the Reviewer #1: for very fruitful comments We have carefully revised the manuscript.

Comment 1. The figures in the manuscript are seen as blurry. The image quality of the figures in the manuscript should be clarified.

Response to Comment 1: We enhanced resolution of the figures.

Comment 2.  The subject transitions in the manuscript should be written more fluently.

Response to Comment 2: Thanks for your valuable comment. We revised it.

Comment 3. The words such as "designed for", "biosensors depends" "been used", "a gold", " throughs howing", “Fe3O4 NPs oxidized”,... is written incorrectly. There are too many typos in the manuscript. The typos in the manuscript should be corrected.

Response to Comment 3: Thanks for your valuable comment. We revised it.

Comment 4.  On page 1, lines 29-30, appropriate references should be added to the sentence in the introduction section.

Response to Comment 4: Thanks for comment. We revised it.

Comment 5.  Figure 1 (especially Figure 1A), should be presented more clearly with necessary and informative information.

Response to Comment 5: Thanks for comment. We revised it.

Comment 6.  Explanations of abbreviations like “TMB molecule, TMBox, Ag” should be added in the manuscript.

Response to Comment 6: Thanks for comment. 3,3′,5,5′-Tetramethylbenzidine or TMB is a chromogenic substrate. Au is gold element symbol and Ag is silver element symbol. We revised it.

Reviewer 2 Report

This review aims to present some news related to the advance in the identification of pathogenic microorganisms by colorimetric diagnostic tests using various techniques

Although the title of the review is recent advances, the novelties in the field of the respective methods are not presented punctually, the article is limited to the present various colorimetric tests, usually starting from their introduction a long time ago. The article is kept in a general and expository manner, without clearly pointing out the novelties that have appeared in recent years. Moreover, the bibliography is under the year 2019, with the exception of self-citations from the year 2020. Regarding the self-citations, the use of pH indicators from natural extracts is presented as a novelty, which, according to the citation, will protect the "user" of the test: "Existing tests that can detect these pH changes use bromthymol blue, phenol red or bromcresol violet as pH indicators. These indicators are synthesized by chemical processes under laboratory conditions and may cause irritation in contact with the eyes and skin. It has both corrosive and toxic properties and may cause irritation in the respiratory system if inhaled. For this reason, inhalation in vapor or gas form should be avoided. Respiratory system organs and kidneys are among the target organs. It is recommended to use protective materials to prevent contact with hands, face, and eyes,,

However, the user has no way to inhale the vapors of an indicator substance impregnated on support.

Also, the current situation regarding the use, in medical practice, of these tests, is not presented. In this context, it would have been useful to compare the use of these tests in combination with other tests, within clear protocols for the detection of infectious diseases and microbial resistance.

The quantitative side of the methods is not evaluated and there is no updated classification regarding the microbial strain and the attached colorimetric test, with the presentation of some qualitative and quantitative characteristics. The article makes many references to the need for cheap techniques but does not offer any economic comparison between the estimated prices of the different detection methods.

Author Response

A response to the comments made by the editors the reviewer

Submission ID: diagnostics-2445497

Title: Recent Advances in Colorimetric Tests for the Detection of Infectious
Diseases and Antimicrobial Resistance

Journal: Diagnostics

Reviewer #2: This review aims to present some news related to the advance in the identification of pathogenic microorganisms by colorimetric diagnostic tests using various techniques.

Response to Reviewer #2: We are very grateful to the Reviewer #2 for very fruitful comments We have carefully revised the manuscript.

Comment 1.  Although the title of the review is recent advances, the novelties in the field of the respective methods are not presented punctually, the article is limited to the present various colorimetric tests, usually starting from their introduction a long time ago. The article is kept in a general and expository manner, without clearly pointing out the novelties that have appeared in recent years. Moreover, the bibliography is under the year 2019, with the exception of self-citations from the year 2020.

Response to Comment 1: Thanks for comment. We revised it.

Comment 2.  Regarding the self-citations, the use of pH indicators from natural extracts is presented as a novelty, which, according to the citation, will protect the "user" of the test: "Existing tests that can detect these pH changes use bromthymol blue, phenol red or bromcresol violet as pH indicators. These indicators are synthesized by chemical processes under laboratory conditions and may cause irritation in contact with the eyes and skin. It has both corrosive and toxic properties and may cause irritation in the respiratory system if inhaled. For this reason, inhalation in vapor or gas form should be avoided. Respiratory system organs and kidneys are among the target organs. It is recommended to use protective materials to prevent contact with hands, face, and eyes. However, the user has no way to inhale the vapors of an indicator substance impregnated on support.

Response to Comment 2: Thanks for comment. There are various toxicity risks in case of direct ingestion, skin contact or inhalation of the mentioned indicators. In all these respects, this is emphasized because anthocyanins are completely biocompatible. In addition, various chromogenic media are in powder form and situations such as inhalation can be observed during the preparation steps. Not all tests may be adsorbed to a solid material.

Comment 3.  Also, the current situation regarding the use, in medical practice, of these tests, is not presented. In this context, it would have been useful to compare the use of these tests in combination with other tests, within clear protocols for the detection of infectious diseases and microbial resistance.

Response to Comment 3: Thanks for comment. We revised it.

Comment 4. The quantitative side of the methods is not evaluated and there is no updated classification regarding the microbial strain and the attached colorimetric test, with the presentation of some qualitative and quantitative characteristics. The article makes many references to the need for cheap techniques but does not offer any economic comparison between the estimated prices of the different detection methods.

Response to Comment 4: Thanks for comment. We revised it.

Round 2

Reviewer 2 Report

The authors of the article have shown openness and cooperation in the direction of increasing the scientific and practical value of the work. They made the suggested changes in a consistent and punctual manner.